# Long COVID, the Brain, Nerves, and Cognitive Function

Allison B. Reiss [1,*], Caitriona Greene [1], Christopher Dayaramani [1], Steven H. Rauchman [2], Mark M. Stecker [2], Joshua De Leon [1] and Aaron Pinkhasov [1]

[1] Department of Medicine and Biomedical Research Institute, NYU Long Island School of Medicine, Long Island, NY 11501, USA; cgree11@u.rochester.edu (C.G.); cdayaramani@gmail.com (C.D.); joshua.deleon@nyulangone.org (J.D.L.); aron.pinkhasov@nyulangone.org (A.P.)

[2] Fresno Institute of Neuroscience, Fresno, CA 93730, USA; dr.rauchman@yahoo.com (S.H.R.); mmstecker@gmail.com (M.M.S.)

* Correspondence: allison.reiss@nyulangone.org

**Abstract:** SARS-CoV-2, a single-stranded RNA coronavirus, causes an illness known as coronavirus disease 2019 (COVID-19). Long-term complications are an increasing issue in patients who have been infected with COVID-19 and may be a result of viral-associated systemic and central nervous system inflammation or may arise from a virus-induced hypercoagulable state. COVID-19 may incite changes in brain function with a wide range of lingering symptoms. Patients often experience fatigue and may note brain fog, sensorimotor symptoms, and sleep disturbances. Prolonged neurological and neuropsychiatric symptoms are prevalent and can interfere substantially in everyday life, leading to a massive public health concern. The mechanistic pathways by which SARS-CoV-2 infection causes neurological sequelae are an important subject of ongoing research. Inflammation- induced blood-brain barrier permeability or viral neuro-invasion and direct nerve damage may be involved. Though the mechanisms are uncertain, the resulting symptoms have been documented from numerous patient reports and studies. This review examines the constellation and spectrum of nervous system symptoms seen in long COVID and incorporates information on the prevalence of these symptoms, contributing factors, and typical course. Although treatment options are generally lacking, potential therapeutic approaches for alleviating symptoms and improving quality of life are explored.

**Keywords:** COVID-19; long COVID syndrome; brain fog; memory; neuroinflammation; neuron; neurologic sequelae

## 1. Introduction

Up to 25% of patients who have recovered from infection with severe acute respiratory syndrome coronavirus 2 (SARS-CoV-2) will experience persistent symptoms, known as long COVID [1–4]. While greater COVID-19 disease severity is correlated with higher risk of long COVID, long COVID can occur irrespective of the initial disease severity [5]. Long COVID, also known as post-acute sequelae of SARS-CoV-2 infection (PASC), refers to the persistence of symptoms for at least 12 weeks after the acute phase of infection [6,7]. Long COVID often entails life-altering neurologic complications [8–10]. Among the most common manifestations affecting mental functioning are impaired thinking ("brain fog"), memory problems, fatigue, sleep disturbances, and headaches [11,12].

In this review, we analyzed the available data from peer-reviewed publications on the neurological and neuropsychiatric symptoms of COVID-19. We discuss the latest research on the detrimental effects of long COVID on cognitive function as well as the underlying mechanisms and potential treatments with a focus on both objective measures, such as neurocognitive testing, blood biomarkers of inflammation and imaging, as well as subjective patient experience.

## 2. Mechanisms Underlying COVID-19 Effects on the Brain

Systemic inflammation and the accompanying elevated production of cytokines and reactive oxygen species are major stressors that, while indirect, can cause pathological effects on the brain (Figure 1) [13–16]. Cytokines can cross even the intact BBB and the barrier becomes more porous under inflammatory conditions; therefore, the brain receives exposure to the elevated cytokine levels that result from COVID-19 infection [17–20]. BBB permeability is increased in human brain tissue obtained from deceased COVID-19 patients [21,22]. Lee et al. showed in an autopsy study of the brains of COVID-19-infected patients that immune-mediated inflammation evidenced by immunoglobulin deposition on the endothelium led to damage of endothelial cells with vascular leakage and loss of vascular integrity [23]. They also found activation of microglia, the innate immune cells of the CNS, and focal areas of platelet aggregation. However, this study did not detect COVID-19 virus in brain tissue. Moreover, reactive microglia can affect oligodendrocytes, leading to impaired myelination which, in mouse models, affects neural function [24]. Furthermore, infection with SARS-CoV-2 can provoke the production of autoantibodies that cross-react with brain tissue and it has been postulated that this autoimmune response could initiate a cycle of structural damage [25,26]. Entry of peripheral leukocytes through the BBB is also facilitated in an inflammatory environment and these cells may themselves be COVID-19-infected, release cytokines, and activate microglia [27].

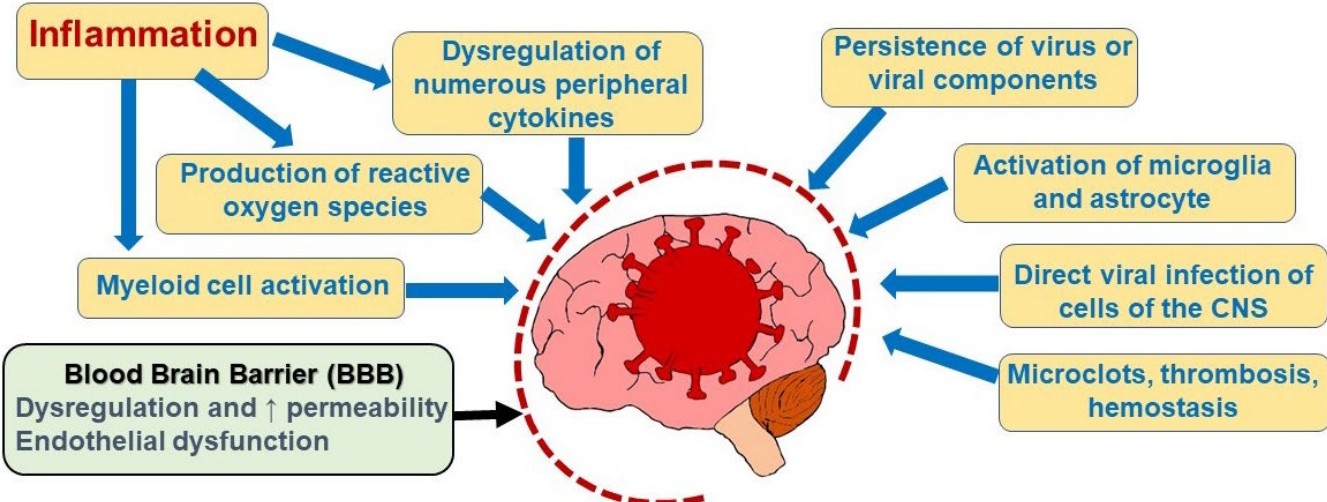

**Figure 1.** Possible mechanisms underlying neurologic symptoms in long COVID. Multiple factors are postulated to contribute to neurologic manifestations of long COVID. Persistent systemic inflammation leads to cytokine production, immune system activation, and production of reactive oxygen species. Increased blood-brain barrier (BBB) permeability allows cytokines to penetrate the brain and induce neuroinflammation. A more porous BBB may also permit direct viral invasion of the brain. Tissue hypoxia may occur due to microclot formation. ↑ = increased.

The COVID-19 virus can directly infect cultured human brain microvascular endothelium [28]. Direct invasion of microvascular endothelium by COVID-19 can weaken the BBB and exacerbate the inflammatory response [28–30]. Inflammation impacts the brain through activation of microglia and astrocytes, which then can dysregulate autophagy and interfere with neurotransmitter production [31–35]. Persistence of viral antigens may play a role in chronic immune system activation and ongoing symptoms [36].

Changes in brain function may be caused not only by the hyperinflammatory environment induced by the virus, but also by direct viral invasion of neurons. SARS-CoV-2 can infect vascular endothelial cells and then may cross into the brain transcellularly through the BBB endothelium [37–39]. Whether the virus replicates robustly in the vascular endothelium is unresolved with conflicting data [40,41]. SARS-CoV-2 has been detected in human brain tissue and has been found in the cerebrospinal fluid of human patients, establishing

its penetration of the BBB into the central nervous system (CNS) [42,43]. In hamsters and mice as well as human organoid models, further evidence supports the potential for SARS-CoV-2 to cross the BBB and infect neurons [44,45]. Post-mortem studies in humans show that the COVID-19 virus can enter the brain, but viral invasion is not the primary cause of neurologic sequelae [46].

An alternate theory is that the virus directly infects olfactory receptor neurons and reaches the brain through the olfactory bulb [47]. The ACE2 receptor is expressed in these neuronal bodies, possibly permitting the infection to move along the olfactory nerve [48,49]. Neuroinvasion by the virus does not generally cause massive spread or replication [50].

The precise contribution of persistent systemic or neuroinflammatory response versus viral invasion of neurons to the development of neurologic and neuropsychiatric symptoms in COVID-19 is still under investigation [24,51]. Emerging evidence suggests that direct neural infection plays a secondary role, while dysregulation of immune-inflammatory pathways plays a more significant role in the development of neurologic and neuropsychiatric symptoms [52]. A summary of the modes through which COVID-19 inflicts damage to the brain and nervous system can be found in Table 1. Environmental and lifestyle disruptions also likely contributed to deteriorating mental health, especially in the face of a worldwide pandemic that resulted in isolation, lack of access to healthcare, and drastic changes in everyday existence on a massive scale for a protracted period of time.

**Table 1.** Neuropathological mechanisms of SARS-CoV-2 long-term effects.

| Mechanism | Cellular and Molecular Changes | References |
|---|---|---|
| Cytokines and leukocytes cross the BBB | Microglial activation, production of neuroinflammatory mediators | [17–20,23,24] |
| Direct viral invasion of microvascular endothelium of the blood-brain barrier | Impaired blood flow in the brain, unclear whether virus enters brain parenchyma via infected endothelium | [28–30,40,41] |
| Entry of viral particles into the brain via the nasal epithelium and olfactory bulb | Neurotoxicity and neuronal loss | [47–49] |

BBB = blood-brain barrier.

Many of the symptoms of long COVID are shared by other disease processes [53,54]. Certainly, other viral infections particularly parvovirus B19 and Epstein–Barr virus, are known to cause myalgic encephalomyelitis/chronic fatigue syndrome (ME/CFS) [55–57]. Lyme disease and rheumatologic diseases, such as lupus and inflammatory arthritis may also be associated with ME/CFS [58–60]. Clinically, some of these symptoms are shared by patients with mild traumatic brain injury, particularly the attentional deficits, fatigue, and pain [61,62].

This leads to a hypothesis that, although injuries to the nervous system may occur through multiple different mechanisms in different diseases, there may be a final common clinical pathway for relatively mild injury that produces the symptoms seen in: Long COVID, chronic fatigue, myalgic encephalomyelitis, and fibromyalgia. Teodoro proposed that reduced externally directed attention due to injury or pain could cause the clinical symptoms and is responsible for the suggested overlap between syndromes [61,63].

This is a topic of future importance and may be addressable by creating a large database not only of long COVID, but also the other diseases discussed above to explore common and distinct symptoms [64]. Combining this clinical information with molecular and imaging markers will help in clarifying the pathophysiology.

## 3. Symptoms of Long COVID

### 3.1. Fatigue

Fatigue is considered a fundamental core symptom of long COVID and occurs after infection with many other viruses [65–69]. This symptom has been reported in a third or more of COVID-19 patients and commonly persists for upwards of 6 months and is an

indicator of worse prognosis [70–72]. Stefanou et al. conducted a longitudinal analysis of 1733 acute COVID-19 patients and found that, at 6 months, 63% had fatigue or muscle weakness [73]. Fatigue is a somewhat subjective experience that is not easily quantified. In long COVID patients, fatigue has been defined as an energy deficit that may be physical, mental, and/or emotional that makes normal daily activities difficult, frequently leaving the patient with post-exertional malaise [74,75]. The acute COVID-19 illness may not have been severe, but ramifications, such as intractable fatigue can be profound.

A qualitative study by Ladds et al. explored the experience of fatigue as described subjectively by patients [76]. They recount a need to adjust their performance of basic activities and disruption of work life with a major decline in functional status due to exhaustion. Self-reported fatigue is associated with impaired quality of life after COVID-19 and this link was also found in studies where fatigue was assessed using more quantitative screening tools [77–79].

The decrease in physical and/or mental performance that results from fatigue may be traced back to changes in CNS provoked by COVID-19 infection and postulated to be a result of both systemic and neuro-inflammatory processes within the brain itself [80,81]. Systemic inflammation and surging cytokine levels can cause or exacerbate tiredness [82].

Diminished neurotransmitter levels in the CNS post-COVID-19 may be responsible for at least some of the fatigue [62,83,84]. A study of 12 post-COVID-19 patients who had recovered from severe pneumonia, but had sustained profound fatigue and 10 healthy controls found neurophysiological indications of disruption of the primary inhibitory neurotransmitter GABA, with evidence of overall reduced GABAergic cortical activity in the post-COVID-19 group [85]. Depleted levels of serotonin may also contribute to fatigue [86,87].

Neuropsychological factors that can contribute to fatigue include anxiety, confusion, depression, apathy, and anger [88,89]. Neuropsychiatric aspects of long COVID are covered in greater detail in the next section.

Nevertheless, another factor that may contribute to the experience of fatigue is the effect of COVID-19 on skeletal muscle which is vulnerable to the ACE2 surface protein [83,90]. Patients may note muscle pain and muscle weakness which limit endurance [91].

*3.2. Neuropsychiatric Sequelae*

The most common long-term neuropsychiatric manifestations of COVID-19 are anxiety, PTSD, and depression and may include pain disorder, delirium, mood swings, and, at the extreme, psychosis [92–95]. Anxiety and depression symptoms have been positively correlated with COVID-19 disease severity and decline in function post-COVID-19 [96]. Alghamdi et al. corroborated these findings with an online survey of 2218 COVID-19 patients finding mood alteration and depression to be common symptoms, which were positively correlated with female sex and disease severity [97]. Recovery is possible as percentage reporting depression decreased over time, but for many patients, symptoms persist for a year and beyond [98].

As with fatigue, inflammatory cytokines are thought to play a pathophysiological role in COVID-19-related depression [99,100]. In a retrospective cohort study of 236,379 patients conducted by Taquet et al., 17.4% were diagnosed with anxiety disorder and 13.7% with a mood disorder in the 6 months following a COVID-19 diagnosis [101]. Long COVID may cause metabolic dysregulation, including the new onset of insulin resistance [102]. Al-Hakeim et al. found an association between insulin resistance and depression in long COVID patients, which they link to the neurotoxicity of oxidative stress in an insulin-resistant milieu [103].

External circumstances, such as isolation, extended quarantines, financial distress, and the stress inflicted by living through the pandemic have all been documented to raise anxiety, incite behavioral changes, increase loneliness, and provoke avoidance behaviors [104–106]. Adding to these environmental factors are the physical changes in permeability of the BBB discussed previously which lead to cytokine overload, inflamma-

tion, and direct viral neuronal invasion with subsequent CNS damage that may mediate neuropsychiatric sequelae [107].

Obsessive-compulsive disorder (OCD) has been reported in many studies with up to 20% of screened patients experiencing symptoms at follow-up [108,109]. OCD symptoms may worsen in persons who already have the disorder, possibly due to the added stressors of masking, hygiene, and isolation and may also appear in those who have not had the diagnosis previously [110–112].

For some patients, neuropsychological symptoms are accompanied by PTSD with potentially debilitating flashbacks, hyperarousal, and intrusive thoughts [113–115]. PTSD occurs in both hospitalized and non-hospitalized patients. In a cohort of 238 patients who were hospitalized in Italy with COVID-19, 17% had PTSD at 4 months post-discharge as assessed by the Impact of Event Scale-Revised [116]. In a study from the Netherlands, Houben et al. found that among 239 patients (62 hospitalized, 177 not hospitalized), PTSD symptoms at 3 months of follow-up were found in 43.5% of patients who had been hospitalized versus 35% of those who had not been hospitalized ($p = 0.23$) while at 6 months of follow-up PTSD symptoms were found in 30.6% of patients who had been hospitalized versus 25.4% of those who had not been hospitalized ($p = 0.42$) [117]. Savarraj et al. found an association between pain and PTSD in a prospective study of hospitalized COVID-19 patients in Texas. Patients who were experiencing pain were seven times more likely to have PTSD at 3 months after hospitalization [118].

Psychosis was also found at higher rates in COVID-19 cohorts than in controls [119,120]. Though a relatively uncommon neuropsychological symptom, multiple case studies have reported patients with sudden onset psychosis both with and without prior medical history after presenting with SARS-CoV-2 [121]. An analysis from Smith et al. of 2396 papers found 48 patients with psychosis lasting between 2 and 90 days, most commonly experiencing delusions [122].

Delirium has also been noted in some COVID-19 patients, especially in older persons and those who are hypoxic or have high fever [123,124]. A study of 516 patients across four Italian medical centers found 73 patients presenting with delirium on admission. Delirium was found to correlate to older age and in-hospital mortality [125].

### 3.3. Sleep Disorders

Among the most commonly reported neurological long COVID symptoms are sleep disturbances [126]. In a study by Huang et al., of 1733 patients suffering from long COVID symptoms, 26% had sleep disturbances [127]. In another study on 251 survivors, 41.8% experienced insomnia at 1 month post-discharge and at 3 months 25.5% still had insomnia. It is estimated that half of patients, even months after acute COVID-19 infection, report sleep-related problems. There is also a bidirectional association between mental health problems and sleep disturbance which may contribute to the mental health complications related to COVID-19 [128]. Patients have reported both trouble sleeping, nightmares and lucid dreaming, which may be a long COVID symptom or a reflection of the stress of life-altering pandemic circumstances [129].

### 3.4. Sensorimotor Deficits

3.4.1. Prevalence and Spectrum of Symptoms

Sensorimotor symptoms of COVID-19 can take a number of forms, including peripheral neuropathy, paresthesias, neuropathic pain, myalgia, and persistent weakness [130–133] (Figure 2). Pilotto et al. found that, at the 6 month follow-up appointment, 40% of previously hospitalized COVID-19 patients had neurologic deficits and that 7.6% of these survivors had subtle motor or sensory deficits [134]. However, an online survey of 3762 patients with COVID-19 from multiple countries found that in the initial 6 months following acute infection, sensorimotor deficits were among the most commonly reported symptoms (91%), exceeding the percentage reporting emotional/mood disorders (88%), headache (77%), and smell/taste disorders (58%). The same study revealed that 55.7% of patients experienced

those symptoms for at least 6 months and that 53.7% were still experiencing those symptoms after 6 months [71]. There are many factors that could be contributing to the difference in numbers seen in these publications, most prominently the variability in defining the spectrum of sensorimotor symptoms as well as the method of collecting data, but it is clear that more work needs to be carried out to assess accurately the prevalence of sensorimotor symptoms following COVID-19.

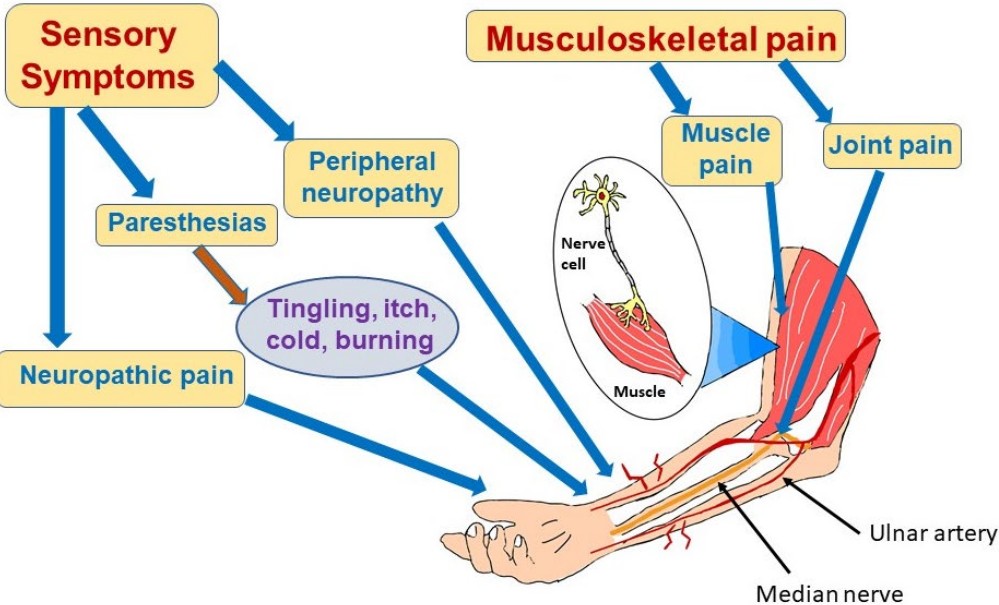

**Figure 2.** Sensorimotor effects of long COVID. Long COVID can cause multiple symptoms in the periphery affecting nerves and muscles as depicted in this figure. Patients may experience nerve pain or paresthesias, most often due to involvement of small nerve fibers. Muscle pain and weakness and joint pain can also be part of long COVID syndrome. Causes of these manifestations are not completely understood, but may result from inflammation and infection-triggered immune system dysregulation. Vasculitis with microclots may also damage nerve and muscle.

3.4.2. COVID-19-Associated Neuropathic Pain and Neuropathies

Neuropathic pain in long COVID patients may involve sensations of itching, tingling, or burning. Although neuropathic pain can have central or peripheral etiologies, neuropathic pain persisting for 3 months after acute COVID-19 infection has been attributed to peripheral neuropathy [135–138]. Although both small and large fiber nerves are affected, recent evidence has shown that it is the small diameter, lightly myelinated or unmyelinated nerves that are most susceptible to damage [139]. The lack of myelination leaves axons subject to local stressors, including those produced by immune dysregulation. Fortunately, these fibers grow continuously throughout a person's lifetime. If the stressful stimulus is removed, reinnervation may occur to a degree sufficient to alleviate symptoms. Although small fiber neurons have been classically thought of as having sensory functions, these nerves are also responsible for innervation of sweat glands, bone, and small blood vessels. Sweat dysfunction has been reported in some post-COVID-19 patients [140]. Interestingly, a small study of 90 patients revealed that patients suffering from neuropathic pain were 4.9 times more likely to have experienced headache during the acute phase of COVID-19 than those suffering from non-neuropathic pain [141].

Paresthesias, experienced as abnormal sensations of tingling, burning, cold, or itch that often occur in the upper or lower extremities, may indicate peripheral neuropathy following COVID-19 infection [142]. A meta-analysis of 36 studies with over 9900 patients found that 33.3% of those with long COVID symptoms reported paresthesias [143]. In agreement with this result, an observational study from Mexico of 280 patients (median age

55) who had been hospitalized with the diagnosis of COVID-19 infection were evaluated up to 6 months after discharge and 35% reported paresthesias [144].

Small fiber peripheral neuropathy may develop within a month of COVID-19 onset [130]. Ser et al. screened patients with a history of COVID-19 infection at least 4 weeks prior to evaluation, and based on an online survey, selected those with high scores in autonomic and neuropathic complaints for further evaluation with electrophysiologic studies [145]. Thirty-eight patients (35.8%) had neuropathic and/or autonomic symptoms and 13 had neuropathic complaints only. The neuropathic symptoms were patchy, mostly proximal, and not symmetrical. An abnormally high cutaneous silent period suppression index ($p = 0.002$) compared to a healthy control group indicated small-fiber dysfunction.

Mononeuropathies that persist have been reported following COVID-19 in many parts of the world [146,147]. New York Presbyterian and Columbia found an association between long COVID and the development of mononeuropathy multiplex [148]. A respiratory clinic in Scotland found elevated hemidiaphragm on chest X-ray in about 3% of patients after COVID-19 pneumonia, likely due to phrenic nerve mononeuritiis. The hemidiaphragm elevation persisted for an average of 7 months after diagnosis of COVID-19 [149].

There has been evidence that COVID-19 infection is associated with demyelinating polyneuropathies, such as Guillain–Barré syndrome (GBS) and Miller–Fisher syndrome [150–152]. Time lapse between COVID-19 onset and symptoms of GBS vary, but may develop in under 2 weeks and generally respond well to standard treatment—either IVIG or plasma exchange [153].

Neuropathy resulting from COVID-19 may be falsely attributed to the state of critical illness seen in some severe acute infections or to compression and traction from prolonged immobility [154]. The treatment options for neuropathy related to COVID-19 are those used for inflammatory neuropathy: Intravenous immunoglobulin (IVIG) and/or corticosteroids [130,131,155]. A short course of steroids is a relatively safe empirical option [156]. Utrero-Rico et al. used prednisone at a dose of 30 mg per day for 4 days while McWilliam used prednisolone at a starting dose of 60 mg per day with tapering over about 8 weeks [157,158]. Dosage of IVIG is generally 2.0 g/kg or higher over a period of 5–7 days, but Thompson et al. used a course of 0.5 g/kg given every 2 weeks, with a plan to continue for between 6 months and 1 year to alleviate symptoms in a small highly subjective study of six long COVID patients [159]. A randomized clinical trial "Immunotherapy for Neurological Post-Acute Sequelae of SARS-CoV-2" is in progress using 0.4 g/kg/day for 5 days versus normal saline with an estimated completion date of April 2024 (NCT05350774).

Gabapentinoids and antidepressants can also be tried [160,161]. Moreover, patients may improve without intervention. COVID-19 can cause a variety of long-lasting sensorimotor symptoms that may not always be reported. Symptoms of neuropathy that linger in long COVID patients are distressing and sometimes disabling and can be difficult to treat pharmacologically [162].

In addition to being an issue for patients as a symptom itself, sensorimotor neuropathy can have profound adverse effects on quality of life. Lasting deficits can make return to work difficult or impossible, cause pain, and impair the ability to perform activities of daily living. The sensorimotor aspect of long COVID is one that may be overlooked, underdiagnosed, and cause lasting problems for patients. More study is needed to grasp the full extent of the problem in order that effective rehabilitation can ensue [163].

### 3.4.3. Myalgias

The long COVID syndrome frequently includes chronic pain commonly in the form of neuropathic pain, but also in the form of myalgias. New onset pain following acute infection with COVID-19 has been seen most frequently in the lower back, the joint space, the neck, and the calf. Risk factors for chronic pain after COVID-19 infection include increasing age and female gender. Older age was positively correlated with the development of non-neuropathic pain [141].

A meta-analysis of over 25,000 COVID-19 patients showed that the prevalence of long COVID myalgias, joint pains, and chest pain ranged from 5.65% to 18.15%, 4.6% to 12.1%, and 7.8% to 23.6%, respectively. Numbers were obtained at onset, as well as 30 days, 60 days, and <180 days after acute infection. The prevalence of musculoskeletal pain decreased between onset and 30 days of infection, increased between 30 and 60 days following infection, and decreased between 60 and <180 days of follow-up [164]. A cohort study at a single center in Turkey showed that, amongst patients with rheumatic or musculoskeletal symptoms after acute infection, at the 3 month follow-up 40.6% had myalgias. Whereas at the 6 month follow-up, only 15.1% had myalgias. Of note, this study also found a significant association between female gender and the development of myalgias following COVID-19 infection [165]. A cross-sectional study from Northern Spain also found female gender to be associated with post-COVID-19 myalgias [166]. Moreover, this study observed that those suffering from post-COVID-19 myalgias had a higher fibrinogen level than those without myalgias (510 $\pm$ 82 mg/dL vs. 394 $\pm$ 87 mg/dL; $p = 0.013$) [166]. A number of studies have found that higher BMI was associated with the persistence of myalgias in the setting of COVID-19 [167,168].

There is limited data on the effective treatment of long COVID myalgias and more work is needed in this area. Physical activity may be helpful in reducing myalgia [169].

### 3.4.4. Pathophysiology of Long COVID Effects on the Peripheral Nerves

The mechanisms by which neurologic damage occurs have yet to be determined definitively, but current theories include invasion of the virus into the nerves directly or indirect effects from toxic processes that change the neural environment. Direct toxicity could occur via invasion of the virus into nerve cells via the angiotensin-converting enzyme 2 (ACE2) receptor or other means, followed by replication and possibly neuronal spread [170,171].

Indirectly, COVID-19 may leave in its wake a milieu of increased cytokine production and release contributing to chronic inflammation and oxidative stress [172]. COVID-19-induced vasculitis may also cause neuropathy since it can lead to microthrombosis within the vasa nervorum [173–175]. A known cause of autoimmune neuropathy seen with other viruses is induction of auto-immunogenicity, possibly by molecular mimicry leading to breaking of self-tolerance. A post-infectious autoimmune cascade could then lead to nerve damage [176].

### 3.5. Cognitive Impairment and Brain Fog

Cognitive deficits are a debilitating symptom experienced between 20 and 35% of patients with post-COVID-19 syndrome following resolution of acute COVID-19 [82,177–180]. Cognitive deficits may be seen in multiple domains compromising concentration, attention, and frontal/executive function [181,182].

A systemic review by Llana et al. of 13 studies of mostly middle-aged adults who had required hospitalization found that one third had subjective cognitive complaints and a highly variable but significant portion had objective deficits in verbal memory at 4–6 months post-COVID-19 [183]. The severity and clinical course of acute COVID-19 infection do not correlate consistently with the appearance or persistence of cognitive symptoms [184,185]. The lack of consistency makes it difficult to predict risk or gain insight into contributing factors and underlying causes of cognitive problems in long COVID patients [186].

Although the term "brain fog" does not have a universally accepted definition, it is a hallmark of long COVID widely used by the lay public and the medical community to describe difficulty thinking and focusing with confusion and lack of mental clarity [107]. Brain fog is generally one part of a symptom cluster, often correlated with decreased psychological and psychomotor performance [187]. A study of 1680 patients aged 18–55 from hospitals in Iran with long COVID symptoms found that 7.2% reported brain fog. Brain fog was positively correlated with factors including female sex, ICU admission, and respiratory problems at the onset of disease [188]. An analysis of retrospective cohort

studies including nearly 1.3 million patients showed that up to 2 years after COVID-19 infection, risk of brain fog continued to be elevated [189]. Other long COVID symptoms (discussed further in other sections), such as fatigue, sleep disturbances, and mood disorders are known contributors to cognitive deficits and may worsen the feeling of brain fog [190]. Neuropsychiatric symptoms, such as depression are connected to cognitive impairments in the realms of global cognition, episodic memory, executive functioning, processing speed, visuospatial memory, attention, and working memory [191]. In a study from Whiteside et al. conducted on 49 patients diagnosed with COVID-19 with self-reported cognitive concerns, neuropsychological tests were administered to observe different areas of cognition: Performance validity, attention/working memory, processing speed, memory, language, visual-spatial, executive functioning, motor, and emotional functioning. Mean scores on objective cognitive measures were not in the impaired range, but there were elevated mean scores for mood measures [192]. The association between depressed mood and brain fog was corroborated in 137 patients in a year-long follow-up after COVID-19 recovery where depression was found to be the strongest predictor of brain fog, leading the authors to suggest that brain fog is a depressive state or the same neuroinflammation is responsible for both symptoms [193]. This study also found that the patients did not have severe cognitive deficits despite brain fog. The link between brain fog and depression is considered an indication that clinical treatment of brain fog would be most effective using a multidisciplinary approach taking neuroinflammation, mental health, sleep quality, stress management, and lifestyle adjustments into account in order to properly address all possible contributing factors [191,194].

While no single pathological hypothesis fully explains brain fog, the presumed etiology is cytokine-mediated during a prolonged immune response in which inflammatory cells and mediators cross the blood-brain barrier, inciting neuroinflammation [195,196]. A study by Nuber-Champier et al. found that higher plasma levels of the inflammatory cytokine tumor necrosis factor (TNF)-$\alpha$ during the acute phase of COVID-19 infection predicted the future risk of memory problems 6–9 months later [197]. He et al. also found a relationship of TNF-$\alpha$ to cognitive deficits even at 15 months after recovery from acute COVID-19 infection [198].

Direct infection of neurons and brain support cells and other mechanisms are also considered as etiologic factors [31,199,200]. Irrespective of causes and objective testing, subjectively, the experience of brain fog is a difficult one for long COVID patients that causes emotional distress and changes in everyday functioning [201].

### 3.6. Hyposmia, Hypogeusia, Hearing Loss

A decline in sensory function has been reported as a symptom associated with long COVID presenting as varying levels of hyposmia (dulled sense of smell), hypogeusia (dulled sense of taste), and hearing loss [73,202]. Although the cause of these symptoms is not fully understood, it is thought that damage to nasal and tongue epithelium due to inflammation as well as viral antigen persistence contribute [203,204]. In relation to smell, olfactory receptor neurons that normally turnover rapidly, exhibit diminished regenerative capability after COVID-19 infection [205]. In a recent meta-analysis, Trott et al. found that about 12.2% of patients experience complete loss of smell (anosmia) and 11.7% lose all sense of taste (ageusia) that continues beyond 12 weeks after COVID-19 infection [206]. A study from Poland conducted from September 2020 to September 2021 of 2218 patients (36.4% female, 63.6% male, mean age 53.8 ± 13.5 years) who had recovered from COVID-19 found that 98 patients (4.4%) reported smell and taste disorders up to 3 months after COVID-19 infection with no difference in the incidence of smell and taste disorders related to disease severity [207]. A study from Wuhan China of 1733 long COVID patients discharged from the hospital between January and May of 2020 found that 11% reported impairment of smell and 7% reported impairment of taste at 6 months [127]. A recent meta-analysis encompassing time-to-event data from 3699 patients in 18 studies utilized self-reported recovery of smell and taste over time after infection to project a likely outcome and predicted

that, similar to the study from Poland, about 5% of patients who had problems with smell and taste initially were likely to suffer persistent dysfunction [208]. Helmsdal et al. performed phone interviews on 170 people who had been diagnosed with COVID-19 between March 2020 and April 2020 in the Faroe Islands and found that by a median of 22.6 months after infection 9% still described symptoms affecting smell and taste [209].

A study from the University of Vienna enrolled 102 patients with COVID-19-related olfactory dysfunction for an in-person evaluation at an average of 216 days after symptom onset. They used not only questionnaires, but also applied chemosensory testing of orthonasal, retronasal, and gustatory function. In this group, recovery proved to occur slowly with only 23.5% returning to normosmia after 216 days. However, only 4% had persistent anosmia, indicating that for most patients, olfactory neurons resume function [210,211]. Some patients who have experienced hyposmia or anosmia as a result of COVID-19 infection also report parosmia, where olfactory response is negatively altered. In one case series, the distortion in smell was reported as reminiscent of sewage, with others reporting rotten meat, rotten eggs, moldy socks, and citrus odors [212]. For the majority of patients, most odors triggered parosmia, but some only experienced this phenomenon for one specific smell, such as perfume, frying smell, or meat. The majority of these patients also experienced dysgeusia, distorted taste. Patients with dysgeusia have described food that was previously appealing as tasting "bland and metallic" [213]. For a significant period after the initial infection, viral presence was found in tongue epithelial cells and taste receptor cells, disrupting taste response. Mucosal inflammation leads to a reduction in epithelial cells and these cells are replenished slowly, causing dysgeusia to be a persistent long COVID symptom [214,215].

Treatment of decreased and distorted sense of smell after COVID-19 infection may encompass olfactory training through exposure to smell essences or oils and odor identification [216,217]. Training can be self-administered or given by a health professional.

Hearing loss is less well-documented after COVID-19 even though it is relatively rare [218,219]. Tinnitus is also reported [220]. In an online survey of over 3700 people, 5.2–6.4% reported hearing loss between months 4–7 after COVID-19 infection [71]. How SARS-CoV-2 affects the auditory pathway is not fully elucidated, but hearing problems may result from epithelial damage and vascular issues, such as microthrombosis [220–222].

Newer variants of COVID-19, such as Delta and Omicron are less likely than the original to cause chemosensory problems. A study by Coelho et al. using a dataset of over 3.5 million cases of COVID-19 found that the probability of smell and taste loss was only 17% for Omicron [223]. The effects of future variants are unknown and the problem may resurge with BA.5. Studies are ongoing to understand the mechanisms through which COVID-19 affects sensory systems and particularly how it may inflict damage to cells that are not specifically infected [224].

### 3.7. Ocular Symptoms

Ocular complications, such as epiphora, hyperemia, and chemosis have occurred in patients who were diagnosed with COVID-19, presumably due to the ACE2 receptors on the cornea, limbus, and conjunctiva. COVID-19 can in turn cause damage to cranial nerves, pupils, lacrimal system, conjunctiva, sclera, retina, choroid, and other parts of the eye [225]. These complications are uncommon, but the virus has been found in tears at low prevalence. In a study performed at a hospital in Turkey, ophthalmologists examined 359 patients hospitalized with a diagnosis of COVID-19 and found that four developed conjunctivitis, five developed subconjunctival hemorrhage, and one experienced vitreous hemorrhage [226]. These complications can develop during infection or at a later time during follow-up. In a study from Egypt, 100 patients who had recovered from an acute COVID-19 infection and 100 control patients who did not have COVID-19 were given ophthalmologic screens. The results of the screening found higher levels of retinal vascular occlusion, uveitis, central serous chorioretinopathy, and anterior ischemic optic neuropathy in those who had been infected with COVID-19 [227]. Retinal microvascular changes may

also be detected after recovery from COVID-19 [228,229]. Endogenous endophthalmitis and ocular surface abnormalities, such as dryness and different tear osmolarity have been reported, as well [230]. Inflammation and elevated coagulation after infection are implicated in significantly higher levels of ocular morbidities due to COVID-19 [229].

## 4. Conclusions

Persistence or appearance of neurologic symptoms after clearance of SARS-CoV-2 infection is a major global health challenge resulting from the COVID-19 pandemic. Adverse effects persisting months after COVID-19 infection can be debilitating and include fatigue, neuropsychiatric sequelae, sleep disturbances, sensorimotor symptoms, cognitive impairment/brain fog, hypoguesia/hyposmia, hearing loss, and ocular symptoms (Table 2). Effective therapies have remained elusive in most cases in these immediate years following the onset of the pandemic. More strategies are needed in order for physicians to effectively treat and manage the long-term neurologic sequelae of COVID-19 infection. However, it is essential for the astute clinician to recognize the chronic neurological manifestations of COVID-19. This knowledge can help in guiding clinical diagnosis and management and ultimately leading to a reduction in unnecessary testing. Further research should bring about improved patient outcomes and satisfaction.

**Table 2.** Neurologic manifestations of long COVID and associated symptoms.

| Neurological Sequelae | Symptoms and Presentation | References |
|---|---|---|
| Fatigue | Physical, mental, or emotional energy deficit that worsens after physical or mental exertion | [65–69,76] |
| Neuropsychiatric | Anxiety, post-traumatic stress disorder, pain disorder, delirium, mood swings, psychosis | [96,97,113–115,119,120] |
| Sleep disturbances | Insomnia, low sleep efficiency, nightmares, lucid dreaming | [126,129] |
| Sensorimotor deficits | Peripheral neuropathy, paresthesias, neuropathic pain, myalgia, persistent weakness | [130–133,141,142] |
| Brain fog | Poor concentration, slowed thinking, difficulty paying attention, and focusing | [181–183] |
| Hyposmia/parosmia | Partial or total loss of sense of smell/misperceiving odors (often pleasant odors seem unpleasant) | [202,206–208,210,212] |
| Hypogeusia/dysgeusia | Partial or total loss of sense of taste/altered perception of taste | [202,206–208,211,213] |
| Hearing problems | Hearing loss, tinnitus | [76,218–220,222] |
| Ocular symptoms | Tearing, hyperemia, chemosis (conjunctival swelling), conjunctivitis, damage to ocular nerves | [225,227] |

Since COVID-19 has affected a large number of patient groups worldwide, we may never fully grasp the impact of long COVID on humanity. The subjective nature of many of the symptoms make them difficult to quantify. Further research is required to better characterize and manage neurologic sequelae in COVID-19 patients. Helping these patients to recover as fully as possible will benefit not only those affected, but also their families and society in general.

**Author Contributions:** Conceptualization, A.B.R., J.D.L. and A.P.; writing—original draft preparation, A.B.R., C.D., C.G. and M.M.S. writing—review and editing, A.B.R., A.P., S.H.R. and M.M.S.; supervision, A.B.R. and S.H.R. All authors have read and agreed to the published version of the manuscript.

**Funding:** This research received no external funding.

**Institutional Review Board Statement:** Not applicable.

**Informed Consent Statement:** Not applicable.

**Data Availability Statement:** Not applicable.

**Acknowledgments:** We thank Lynn Drucker, Edmonds Bafford, and Robert Buescher. Original art by Shelly Gulkarov. In memory of James Gilmore Drayton, loving husband, father, and grandfather.

**Conflicts of Interest:** The authors declare no conflict of interest.

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
