# Peer review of "Long COVID, the Brain, Nerves, and Cognitive Function"

_2035-8377, doi:10.3390/neurolint15030052_

Round 1

Reviewer 1 Report

The manuscript (neurolint-2426204) unambiguously describes long term complications after COVID-19 infection, which may incite changes in brain function with a wide range of lingering symptoms. The authors have well summarized the past COVID effects and have used appropriate references. I have one minor concern about the terminology of COVID-19 as below-

Minor Concern-

1.     Authors are suggested to use consistent terminology for COVID-19 in entire manuscript. As many places they used as “Covid” while many other places as “Covid-19”. COVID-19 is an acronym. In its full form, COVID-19 stands for COrona VIrus Disease of 2019.

Author Response

We thank the reviewer for thoroughly scrutinizing our manuscript. As requested, we have revised the manuscript and addressed the specific comments of the reviewer. The revised sections are delineated in red in a marked copy of the manuscript text.

Below, we provide a point-by-point response to the reviewer’s comments.

Reviewer # 1 Comments

  • COMMENT #1: Authors are suggested to use consistent terminology for COVID-19 in entire manuscript. As many places they used as “Covid” while many other places as “Covid-19”. COVID-19 is an acronym. In its full form, COVID-19 stands for COrona VIrus Disease of 2019..

RESPONSE: We thank the reviewer for the favorable assessment of our manuscript. We apologize for the inconsistency and have corrected it throughout referring to the acute disease as “COVID-19” and the syndrome of post-COVID-19 symptoms as “long COVID” as seems to be the convention in the published literature.

Reviewer 2 Report

This is a timely review on a hot and still largely to be explored topic, the Long Covid and its neurological effects. The authors' viewpoint is clinical, which is fine. However, this implies that a lot of experimental studies on SARS-CoV-2 brain pathophysiology is left out. This means that paragraph 2's contents are mostly generic. They  do not distinguish the source (animal or human) of the experimental data, ignore the role of inflammasomes' activation, the mechanisms by which viral infection (to be preferred to the word  "invasion") of the neural cells, and the roles played by the microglia. I invite the authors to add some more information about these topics to strengthen the scientific value of this paragraph.

Regarding Figure 1, the three mechanisms regarding BBB and endothelial dysfunction should be unified. Moreover, the figure does not include any indication of peripheral nerves and of skeletal muscles, so it is incomplete and somewhat misleading . Finally, the caption should be enriched with more pertinent details .

The clinical part is generally well developed, although there are repetitions about the same symptoms (e.g. fatigue, brain fog, etc.) in the different paragraphs, which were possible should be avoided.

At line 97 the word "overlap" is repeated twice. 

At line 427 the sentence "In a study etc." the verb is missing.

In Table 1 the ocular symptoms are in oversize characters.

The English language is reasonable though some minor editing particularly of lengthy sentences and of some typos would improve it.

Author Response

We thank the reviewer for thoroughly scrutinizing our manuscript. As requested, we have revised the manuscript and addressed the specific comments of reviewer #1. The revised sections are delineated in red in a marked copy of the manuscript text.

Reviewer # 2 Comments

  • COMMENT: The authors' viewpoint is clinical, which is fine. However, this implies that a lot of experimental studies on SARS-CoV-2 brain pathophysiology is left out. This means that paragraph 2's contents are mostly generic. They do not distinguish the source (animal or human) of the experimental data, ignore the role of inflammasomes' activation, the mechanisms by which viral infection (to be preferred to the word "invasion") of the neural cells, and the roles played by the microglia. I invite the authors to add some more information about these topics to strengthen the scientific value of this paragraph.

RESPONSE: We have added more detail to this section with human-specific references.

  • COMMENT: Regarding Figure 1, the three mechanisms regarding BBB and endothelial dysfunction should be unified. Moreover, the figure does not include any indication of peripheral nerves and of skeletal muscles, so it is incomplete and somewhat misleading. Finally, the caption should be enriched with more pertinent details.

RESPONSE: We have remade the figure and elaborated within the caption. We have added a second figure (Figure 2) depicting neuropathy and myalgias.

  • COMMENT: The clinical part is generally well developed, although there are repetitions about the same symptoms (e.g. fatigue, brain fog, etc.) in the different paragraphs, which where possible should be avoided.

RESPONSE: We have done our best to remove repetitious statements.

  • COMMENT: At line 97 the word "overlap" is repeated twice.

RESPONSE: We have corrected this error.

  • COMMENT: At line 427 the sentence "In a study etc." the verb is missing.

RESPONSE: We have corrected this sentence.

  • COMMENT: In Table 1 the ocular symptoms are in oversize characters.

RESPONSE: We have fixed the font size.

Reviewer 3 Report

Dear authors,

Here are the suggested revisions for the mentioned sections:

Introduction:

1. Consider adding a sentence to provide a brief background on the prevalence of long Covid before mentioning the symptoms. For example: "Up to 25% of patients who have recovered from SARS-CoV-2 infection experience persistent symptoms, known as long Covid."

2. Instead of using "regardless of severity," you can use a phrase like "irrespective of the initial disease severity" to make the sentence more precise.

3. Rephrase the sentence "Referred to as 'long Covid', it occurs when symptoms continue to be present at least 12 weeks following acute infection" to enhance readability. For example: "Long Covid, also known as post-acute sequelae of SARS-CoV-2 infection (PASC), refers to the persistence of symptoms for at least 12 weeks after the acute phase of infection."

4. In the last sentence of the introduction, briefly explain what you mean by objective measures, such as neurocognitive tests or imaging techniques, to give readers an idea of the methodologies used in your review.

Mechanism:

1. Include a diagram or table to illustrate the different modes of SARS-CoV-2 invasion in the brain and the associated cellular and molecular alterations. This will help visualize the mechanisms involved, including immune system activation, cytokine release patterns, and their deleterious outcomes.

2. Rewrite the sentence "Whether neurologic and neuropsychiatric symptoms of Covid are due predominantly to a persistent systemic or neuroinflammatory response or to viral invasion of neurons is still unclear and further study is underway [42]" to clarify the meaning. For example: "The precise contribution of persistent systemic or neuroinflammatory response versus viral invasion of neurons to the development of neurologic and neuropsychiatric symptoms in Covid-19 is still under investigation."

3. Rewrite the sentence ""The available evidence suggests that direct neural infection plays a less prominent role, while dysregulation of the immune-inflammatory pathways is more influential in the development of neurologic and neuropsychiatric symptoms [43]" as follows: "Emerging evidence suggests that direct neural infection plays a secondary role, while dysregulation of immune-inflammatory pathways plays a more significant role in the development of neurologic and neuropsychiatric symptoms [43]."

After mechanism section mentioned headings should be part of "Symptoms"

1. Split the sentence "Long Covid may cause metabolic dysregulation, including the new onset of insulin resistance, and Al-Hakeim et al. found an association between insulin resistance and depression in long Covid patients, which they link to the neurotoxicity of oxidative stress in an insulin-resistant milieu [93,94]" into two sentences for clarity and readability.

2. In the sentence "A study of 516 patients across 4 Italian medical centers found 73 presenting with delirium on admission," add "patients" after "73" for grammatical accuracy.

3. After mentioning the online survey, remove the comma after "survey" in the sentence "based on an online survey, selected those with high scores in autonomic and neuropathic complaints for further evaluation with electrophysiologic studies [136]."

4. Add a few lines or provide a tabular representation about the pathogenesis and disease mechanism of hyposmia, hypogeusia, hearing loss, vision problems, cognitive impairment, and brain fog to provide a comprehensive overview of these symptoms.

5. Regarding the treatment options for neuropathy related to Covid-19, provide additional information about the use of IVIg and corticosteroids as part of the treatment approach [121,122,145].

6. In the sentence "This study also observed that those suffering from post-Covid myalgias had a higher fibrinogen level than those without myalgias (510 ± 82 mg/dL vs 394 ± 87; p = .013) [152]." add Unit.

Little correction is required. 

Author Response

We thank the reviewer for thoroughly scrutinizing our manuscript. As requested, we have revised the manuscript and addressed the specific comments of reviewer #1. The revised sections are delineated in red in a marked copy of the manuscript text.

Reviewer # 3 Comments

  • COMMENT: Consider adding a sentence to provide a brief background on the prevalence of long Covid before mentioning the symptoms. For example: "Up to 25% of patients who have recovered from SARS-CoV-2 infection experience persistent symptoms, known as long Covid."

RESPONSE: We have made the suggested change at the beginning of the introduction.

  • COMMENT: Instead of using "regardless of severity," you can use a phrase like "irrespective of the initial disease severity" to make the sentence more precise.

RESPONSE: We have made the suggested change in phrasing.

  • COMMENT: Rephrase the sentence "Referred to as 'long Covid', it occurs when symptoms continue to be present at least 12 weeks following acute infection" to enhance readability. For example: "Long Covid, also known as post-acute sequelae of SARS-CoV-2 infection (PASC), refers to the persistence of symptoms for at least 12 weeks after the acute phase of infection."

RESPONSE: We have made the suggested change in phrasing.

  • COMMENT: In the last sentence of the introduction, briefly explain what you mean by objective measures, such as neurocognitive tests or imaging techniques, to give readers an idea of the methodologies used in your review.

RESPONSE: We have made the suggested addition.

  • COMMENT: Include a diagram or table to illustrate the different modes of SARS-CoV-2 invasion in the brain and the associated cellular and molecular alterations. This will help visualize the mechanisms involved, including immune system activation, cytokine release patterns, and their deleterious outcomes.

RESPONSE: Added as new Table 1.

  • COMMENT: Rewrite the sentence "Whether neurologic and neuropsychiatric symptoms of Covid are due predominantly to a persistent systemic or neuroinflammatory response or to viral invasion of neurons is still unclear and further study is underway [42]" to clarify the meaning. For example: "The precise contribution of persistent systemic or neuroinflammatory response versus viral invasion of neurons to the development of neurologic and neuropsychiatric symptoms in Covid-19 is still under investigation."

RESPONSE: We have made the suggested change in phrasing.

  • COMMENT: Rewrite the sentence ""The available evidence suggests that direct neural infection plays a less prominent role, while dysregulation of the immune-inflammatory pathways is more influential in the development of neurologic and neuropsychiatric symptoms [43]" as follows: "Emerging evidence suggests that direct neural infection plays a secondary role, while dysregulation of immune-inflammatory pathways plays a more significant role in the development of neurologic and neuropsychiatric symptoms [43]."

RESPONSE: We have made the suggested change in phrasing.

  • COMMENT: Split the sentence "Long Covid may cause metabolic dysregulation, including the new onset of insulin resistance, and Al-Hakeim et al. found an association between insulin resistance and depression in long Covid patients, which they link to the neurotoxicity of oxidative stress in an insulin-resistant milieu [93,94]" into two sentences for clarity and readability.

RESPONSE: We have split into 2 sentences.

  • COMMENT: In the sentence "A study of 516 patients across 4 Italian medical centers found 73 presenting with delirium on admission," add "patients" after "73" for grammatical accuracy.

RESPONSE: Done

  • COMMENT: After mentioning the online survey, remove the comma after "survey" in the sentence "based on an online survey, selected those with high scores in autonomic and neuropathic complaints for further evaluation with electrophysiologic studies [136]."

RESPONSE: Done

  • COMMENT: Add a few lines or provide a tabular representation about the pathogenesis and disease mechanism of hyposmia, hypogeusia, hearing loss, vision problems, cognitive impairment, and brain fog to provide a comprehensive overview of these symptoms.

RESPONSE: There is very limited information on mechanism, but we have added the latest available with references 203-205 and 220-222.

12) COMMENT: Regarding the treatment options for neuropathy related to Covid-19, provide additional information about the use of IVIg and corticosteroids as part of the treatment approach [121,122,145].

RESPONSE: We have put in additional information with new references 156-159.

13) COMMENT: In the sentence "This study also observed that those suffering from post-Covid myalgias had a higher fibrinogen level than those without myalgias (510 ± 82 mg/dL vs 394 ± 87; p = .013) [152]." add Unit.

RESPONSE: Done